# Resveratrol inhibits bladder cancer proliferation by targeting the AURKA/STAT3 axis: From computational analysis to experimental validation

**Chao Feng, Guodong Chen, Yan Shu, Jing Chen, Wenxin Ye, Ligang Ren**[ID]*

Department of Urology, Tongde Hospital of Zhejiang province, Hangzhou, Zhejiang, China

* tdwztdzy@163.com

## Abstract

### Introduction

Given the high recurrence rate of bladder cancer (BCa) and the significant adverse effects associated with conventional treatments, it is urgent to search for new clinical therapeutic targets and safer natural-derived compounds. Resveratrol (Res) has been demonstrated to exhibit cytotoxicity against various tumors. However, the signaling pathways and targets involved in inhibition of BCa cells still need further exploration. This study aims to investigate the mechanism of Res in Bca via suppression of the AURKA/STAT3 axis, providing important theoretical basis for subsequent further researches on Res for treating BCa.

### Methods

Differentially expressed genes were identified through bioinformatics methods and the binding sites of resveratrol were also identified. The cell survival rate was detected by the CCK8 method to calculate the concentrations of Res for 30% inhibition and for 50% inhibition. Then, flow cytometry was used to detect the cell cycle and apoptosis after treatment with different concentrations of Res. Immunofluorescence staining was used to detect the effects of Res and MLN8237 on the expression of STAT3. Western blot and qPCR analyses were used to verify the reliability of the effects of Res and MLN8237 on target proteins.

### Results

AURKA was identified as the potential target of Res by computational analysis. Further validation through CCK8 assays and flow cytometry demonstrated that Res could inhibit BCa cells and their cell cycle in a time- and dose-dependent manner. Immunofluorescence staining revealed both Res and MLN8237 suppressed STAT3 expression in BCa cells. Additionally, western blot and qPCR analysis confirmed

**Data availability statement:** All relevant data are within the manuscript and its Supporting Information files.

**Funding:** This work was supported by the Traditional Chinese Medicine Science and Technology Project of Zhejiang Province (2024ZR060), URL: https://zycc.wsjkw.zj.gov.cn/, and CF received this award. The funders did not play any role in the study design, data collection and analysis, decision to publish, or preparation of the manuscript.

**Competing interests:** The authors have declared that no competing interests exist.

that Res and MLN8237 inhibited the expression of AURKA and known target genes (VEGF, Bcl-2, and Cyclin D1).

## Conclusion

Our findings suggest that Res may regulate BCa cell expression through the AURKA/STAT3 axis, providing a theoretical foundation for the structural optimization of Res and the development of multi-target drugs for clinical application.

---

## 1 Introduction

Bladder cancer (BCa) is the sixth most aggressive tumor among males worldwide, and after the 10th place among women. More than 600,000 new BCa cases are diagnosed in 2022 and patients experience a substantial economic burden [1]. At present, traditional main therapeutic methods are still transurethral resection of bladder tumor (TURBT), radical cystectomy (RC), chemotherapy and BCG vaccine [2,3]. Unfortunately, approximately 50% of BCa patients have a poor prognosis due to locoregional relapse (LRR) and distant metastasis, especially in muscle invasive bladder cancer (MIBC) patients [4,5]. Thus, there is a critical need for effective evaluation and timely adjuvant therapeutic strategies [6].

BCa is regarded as a heterogeneous disease and influenced by the regulation of particular target genes [7,8]. Although some new targeted drugs and immunotherapies hold promise for the treatment of BCa at present, high medical costs and adverse reactions are limited wide clinical application [9–11]. Therefore, it is particularly important to search for specific target genes and seek low-toxicity, effective and inexpensive therapeutic drugs.

Resveratrol (Res), a naturally occurring non-flavonoid polyphenolic compound, is widely distributed in plants such as grapes and polygonum. Current studies have confirmed that this compound exhibits significant pharmacological activities such as antioxidant, anti-inflammatory, and anti-tumor effects [12,13]. Res shows significant application potential in the prevention and treatment of BCa due to its wide availability, low cost, and clear tumor suppression effect [14]. In recent years, the molecular mechanism research of its use in BCa treatment has become a hot topic [15,16]. However, the preclinical mechanism studies on the anti-BCa effect of Res are not sufficient. Especially in terms of the identification of its molecular targets and the regulation of related signaling pathways, further in-depth exploration is still needed.

Therefore, this study aims to combine bioinformatics, molecular docking and experimental validation to systematically reveal the potential targets, biological functions and related signaling pathways of Res in the treatment of BCa, as shown in Fig 1. Two hub genes in BCa, AURKA and CDK1, were verified through analyzing the microarray dataset GSE130598 from the Gene Expression Omnibus (GEO) database and gene data patterns of BCa from the TCGA database. Subsequently, molecular docking models of Res-AURKA were constructed using Autodock software and indicated that there were multiple binding sites between AURKA and Res.

**Fig 1. The workflow of computational analysis and experimental validation.** GEO, Gene Expression Omnibus; DEGs, differentially expressed genes; TCGA, The Cancer Genome Atlas; GO, Gene Ontology; PPI, Protein-protein Interaction; GEPIA, Gene Expression Profiling Interactive Analysis; HPA, Human Protein Atlas; qPCR, quantitative Real-time PCR; TIMER, Tumor Immune Estimation Resource; CCK8, Cell Counting Kit 8.

Finally, through in vitro experiments, AURKA further confirmed as a potential target of BCa and Res may regulate BCa cell expression through the AURKA/STAT3 axis. These results will provide a basis for the development and application of Res drugs in BCa, and also have referential significance to the subsequent research on natural drugs for treating BCa.

## 2 Materials and methods

### 2.1 Data collection and preprocessing

A workflow chart illustrating this study is shown in Fig 1. The microarray dataset GSE130598 and relevant clinical data such as age, sex and disease stage were filtered from the GEO database (https://www.ncbi.nlm.nih.gov/geo/) and analyzed using the GPL26612 platform. GSE130598 contained 48 samples from 24 patients. Data lacking relevant clinical information was screened out and then we selected 32 samples, which included 16 MIBC and 16 healthy bladder samples, were used to verify the differentially expressed genes (DEGs). The gene expression profiles of MIBC and normal patients were also collected from the TCGA database (https://portal.gdc.cancer.gov/). The "limma" packages in R software (version 4.0.5) were used to ensure the validity of the data. A total of 132 MIBC and 2 normal bladder samples were filtered out. These data were used to identify immune cell infiltration signatures between MIBC and normal bladder samples. Additionally, the correlation between DEGs and immune infiltration cells (IICs) was examined by the Tumor Immune Estimation Resource (TIMER) database analysis tool. An error detection rate of < 0.05 and a |log2 Fold Change| (|log2FC|) of ≥1 was used as the threshold for screening the DEGs.

### 2.2 Functional enrichment analysis and Protein-protein interaction (PPI) network construction

Gene Ontology (GO), a major bioinformatics tool, was used to analyze the relevant biological processes from different dimensions and levels. Three major parts were conducted. GO analysis was utilized based on the "cluster profiler" and "enrich plot" package, and an adjusted-$p$ value < 0.05 was regarded as the screening criterion. Using the DEGs identified above, the Search Tool for the Retrieval of Interacting Genes (STRING) was utilized to establish a PPI network to further study their function at the protein level. Cytoscape software (version 3.6.1) was used to visualize the PPI networks and based on the connected gene nodes, ten hub genes were selected for further analysis and their relative expression was determined. Hub genes refer to nodes representing proteins that exhibit the high connectivity and they play critical roles in coordinating biological processes and serve as key regulators within the network.

## 2.3 Differential expression analysis of the hub genes

Hub genes were further assessed using Gene Expression Profiling Interactive Analysis (GEPIA), an online database that facilitates the standardized analysis of RNA-seq data (http://gepia.cancer-pku.cn/index.html). The expression level of specific DEGs was analyzed using the MIBC, matched TCGA normal and Genotype-Tissue Expression (GTEx) data. $|log2FC| \geq 1$ and adjusted-$p < 0.05$ were considered statistically significant.

## 2.4 Correlation between the immune cell infiltration profile and DEGs

A total of 22 IIC types were collected and analyzed using the CIBERSORT algorithm. In this study, an adjusted-$p < 0.05$ was defined as the evaluation criterion required to assess the score results of related immune cell groups. After filtration, the data was used for further processing. The Wilcoxon rank-sum test was used to compare the infiltration levels of different immune cell types between MIBC and normal control samples. Survival analysis of specific immune cell subtypes was performed based on the median of various immune cell proportions. "Survival" packages were used to compare the survival rates of patients with high and low expression of immune cells.

## 2.5 Protein expression of the hub genes and tumor-infiltrating immune cell analysis

Protein expression of the hub genes in MIBC samples was verified using Human Protein Atlas (HPA, https://www.proteinatlas.org) data. The selected results of the hub genes were examined using the Tumor Immune Estimation Resource (TIMER, https://cistrome.shinyapps.io/timer/) database, which was used for systematic analysis of immune infiltrations with six usual types of IICs. The correlation between expression of the screened hub genes and the six IIC types was determined and visualized.

## 2.6 Real time qualitative PCR and computational molecular simulation docking model

The relative mRNA expression levels of ABL1, CDK1, FYN and AURKA between BCa cell UMUC-3 and normal bladder epithelial cell SV-HUC-1 were detected by PCR. The UMUC-3 cells (CL-0463, Procell, Wuhan, China) were cultured in complete MEM medium (KGM41500S-500, Keygen BIO, Jiangsu, China) and the SV-HUC-1 cells (IM-H068, Immocell, Xiamen, China) were cultured in a dedicated medium (IM-H068-1, Immocell, Xiamen, China). These cells were cultured in an incubator with 5% carbon dioxide, and the temperature was maintained at 37°C. The PCR primers for the four hub genes (ABL1, AURKA, CDK1, and FYN) were designed using Primer 5 software. RNA was extracted from the cells using Trizon reagent (CW0580S, Cwbio, Jiangsu, China), and 0.2 mL chloroform was added for each 1 mL of Trizon. mRNA was extracted using the mRNA ultrapure extraction kit (CW0581M, Cwbio, Jiangsu, China), and miRNA was extracted using the miRNA extraction kit (CW0627S, Cwbio, Jiangsu, China). The OD260/280 ratio of mRNA and miRNA was determined using a UV-visible spectrophotometer. cDNA was synthesized using HiScript II Q RT SuperMix for qPCR (+gDNA wiper) (R223-01, Vazyme, Nanjing, China), and fluorescence quantitative PCR was performed using a fluorescence PCR instrument. The forward and reverse primers were as follows: β-actin (TGGCACCCAGCACAATGAA and CTAAGTCATAGTCCGCCTAGAAGCA); AURKA (CACCCAAAAGAGCAAGCAGC and TTACCCAGAGGGCGACCAA); ABL1 (TGAAGCCGCTCGTTGGA and AGACTGTTGACTGGCGTGATGT); CDK1 (GGGGTCAGCTCGTTACTCAAC and GCCCAAAGCTCTGAAAATCCTG); FYN (TACCAAATCTTGTGGACATGGCA and GATGGGGAACTTTGCACCTTG).

Molecular docking of Res and AURKA proteins was conducted using Autodock software to further analyze their interaction. Firstly, the 3D structure of AURKA was selected from the Protein Data Bank (PDB) database (https://www.rcsb.org/). Due to the excessive flexibility of some loops in the protein which might lead to low electron cloud density and miss coordinate information, the missing amino acids of the AURKA structure were reasonably reconstructed through the AlphaFold3 database (https://alphafold.ebi.ac.uk/) before analyzing its 3D spatial structure. Subsequently, the 3D structure of the Res ligand was obtained from the PubChem website (https://

pubchem.ncbi.nlm.nih.gov/). The selected AURKA receptor and Res ligand were imported into PyMol software, and the system automatically performed some preprocessing such as removing water molecules and metal ions to simplify the calculation. Then, the number of poses were selected on the docking page, and the default active pocket region of AURKA was 10. The substrate molecule was docked into the active pocket region through semi-flexible docking, resulting in 10 docking results. The docking model with the lowest energy was selected as the final docking result.

## 2.7 Cell counting kit 8 (CCK8) and flow cytometry detection

10 mg of resveratrol powder (HY-16561, MCE) was dissolved in 2.19 ml of dimethyl sulfoxide (DMSO, Solarbio, Beijing, China) to form a 20 mM stock solution, which was then diluted with MEM complete medium to 0, 400, 600, 800 and 1000 µM for treating UMUC-3 cells for 2 hours, followed by treatment with 0, 50, 100, 150 and 200 µM for 24 hours. The AURKA inhibitor MLN8237 solution (HY-10971, MCE) was prepared into 0, 2, 10, 50, 100, and 150 µM with MEM complete medium to treat UMUC-3 cells for 24 hours. Subsequently, the cells to be tested were collected with a volume of 100 µl per well. 10 µl of CCK8 reagent (KGA317, Keygen BIO, Jiangsu, China) was added to each well, and they were incubated in the incubator for 2 hours. The absorbance of each well was detected at 450 nm with an enzyme marker (SuPer Max3100, Shanghai, China). The cell survival rate was detected by the CCK8 method to calculate the concentrations of Res for 30% inhibition (IC30) and for 50% inhibition (IC50). At the same time, the half-maximal inhibitory concentration of MLN8237 was calculated.

According to the different concentrations of Res, the cells were divided into 0 concentration, IC30 concentration, IC50 concentration and high concentration groups. After treating the cells of the four groups for 2 hours and 24 hours, $1 \times 10^6$ cells were collected from each group. The cells were washed twice with 1 ml PBS (G4202, Servicebio, Wuhan, China) at 1500 rpm and the supernatant was discarded. Then, 500 µl of apoptosis positive control solution was added to resuspend the cells, which were incubated on ice for 30 minutes. After adding 1 ml PBS for washing, an appropriate amount of $1 \times$ binding buffer was added to resuspend the cells and an equal number of untreated cells were mixed with them. Each sample was divided into 3 tubes of 1.5 ml, and 5 µl Annexin V-FITC and 10 µl PI (AP101–100-kit, MULTI SCIENCES, Hangzhou, China) were added to each tube and gently mixed. After incubation at room temperature for 5 minutes, the apoptosis of the cells was detected by flow cytometry (NovoCyte 2060R, Aisen Gene, China). Next, we used the Cell Cycle Staining Kit (CCS102, MULTI SCIENCES, Hangzhou, China) to detect the cell cycle of the above groups of cells. $1 \times 10^6$ cells were collected from each group. The cell suspension was washed by centrifugation at 1500 rpm and the supernatant was discarded. Then, 1 ml PBS was added and centrifuged at 1500 rpm to discard the supernatant. Subsequently, 1 ml DNA staining solution and 10 µl permeabilization solution were added, vortexed for 5–10 seconds to mix, and incubated at room temperature in the dark for 30 minutes. The cell cycle was detected by flow cytometry (NovoCyte 2060R, Aisen Gene, China).

## 2.8 Immunofluorescence staining and detection of STAT3 pathway genes

UMUC-3 cells were divided into 0-concentration control group, IC50 Res group, and IC50 MLN8237 group. After 24 hours of culture, the culture dishes of each group were fixed with 4% tissue cell fixation solution (P1110, Solarbio, Beijing, China) for 15 minutes and then washed three times with PBS. After blocking with 5% BSA (A8020, Solarbio, Beijing, China) at 37°C for 30 minutes, the cells were incubated overnight at 4°C with diluted rabbit anti-STAT3 (ET1605−45, HUABIOI, 1/200). After being rewarmed at room temperature for 45 minutes and washed three times with PBS, diluted Cy3 Goat Anti-Rabbit IgG (H + L) (AS007, ABclonal, 1/200) was added and incubated at 37°C for 30 minutes, followed by rinsing with PBS. The specimens were stained with DAPI staining solution (KGA215−50, Keygen BIO, Jiangsu, China), rinsed with PBS, and dried before being sealed. The stained cells were observed and images were collected under a fluorescence microscope (CKX53, Olympus, Janpan).

Further, according to the different reagents added, UMUC-3 cells were divided into blank control group, IC30 Res group, IC50 Res group and IC50 MLN8237 group. After 24 hours of culture, RNA was extracted from the collected cells using the ultra-pure RNA extraction reagent (CW0581M, Cwbio, Jiangsu, China), and cDNA was synthesized using the HiScript II Q RT SuperMix for qPCR (+gDNA wiper) (R223-01, Vazyme, Nanjing, China). Fluorescence quantitative PCR was performed using a fluorescence PCR instrument. The reaction steps were as follows: pre-denaturation at 95°C for 30 seconds; denaturation at 95°C for 10 seconds; annealing at 58°C for 30 seconds; extension at 72°C for 30 seconds; 40 cycles. The relative expression levels of AURKA and three downstream genes of the STAT3 pathway, cyclinD1, VEGF, and BCL-2 mRNA, were calculated according to the 2-△△Ct method. The forward and reverse primers were as follows: β-actin (TGGCACCCAGCACAATGAA and CTAAGTCATAGTCCGCCTAGAAGCA); AURKA (CACCCAAAAGAGCAAG-CAGC and TTACCCAGAGGGCGACCAA); Bcl-2 (GAGGATTGTGGCCTTCTTTG and GCCGGTTCAGGTACTCAGTC); Cyclin D1 (GGCGGAGGAGAACAAACAGA and GGCGGATTGGAAATGAACTT); VEGF (GAGGAGGGCAGAATCATCAC and TGAGGTTTGATCCGCATAATCT).

Western Blot (WB) was used to detect the expression differences of AURKA, cyclinD1, VEGF, and BCL-2 proteins. Firstly, 200 μL of cell RIPA lysis buffer (C1053, Applygen, Beijing, China) was added to each well, and the cells were disrupted using a cell disruptor. The samples were then centrifuged at 12,000 r/min at 4°C for 10 minutes, and the super-natant was collected and transferred to a new centrifuge tube for protein quantification. The protein concentration was quantified using the BCA protein quantification kit (E-BC-K318-M, Elabscience, Wuhan, China). The protein samples were denatured and stored at −20°C. Sodium dodecyl sulfate-polyacrylamide gel electrophoresis (SDS-PAGE) was performed for 1.5 hours, followed by constant current transfer at 300 mA for 1−2 hours. The PVDF membrane (IPVH00010, Milli-pore, USA) was blocked with a special defat dried milk (P1622, Applygen, Beijing, China) and incubated overnight at 4°C. The primary antibodies were: Mouse Anti-GAPDH (HC301, TransGen Biotech, 1/2000) for the internal reference, Mouse Anti AURKA (66757−1-Ig, Proteintech, 1/2000), Mouse Anti Bcl-2 (YM3041, ImmunoWay, 1/1000), Rabbit Anti VEGF (AF5131, Affinity, 1/1000), and Rabbit Anti cyclinD1 (ab134175, abcam, 1/20000). The next day, the PVDF membrane was incubated with the secondary antibodies at room temperature for 2 hours: HRP conjugated Goat Anti-Mouse IgG (H+L) (GB23301, Servicebio, 1/2000) for the internal reference, and HRP conjugated Goat Anti-Rabbit IgG (H+L) (GB23303, Servicebio, 1/2000) for the target. The membrane was washed, soaked in the luminescent solution and imaged using a high-sensitivity chemiluminescence imaging system (Tanon-5200, Tanon, Shanghai, China).

## 2.9 Statistical analysis

GraphPad Prism 9.5 software was used for statistical analysis and graphing. All experiments were repeated three times, and the quantitative results were expressed as mean±standard deviation (X±S). One-way analysis of variance was used for comparison of quantitative values among multiple groups, with a significance level of α=0.05. P value<0.05 was con-sidered statistically significant.

## 3 Results

### 3.1 The identification of DEGs in BCa

The "limma" R package was used to extract differential genes from GSE130598. Of the 182 screened DEGs, 98 were up-regulated and 84 were down-regulated. All regulated genes were summarized and plotted on a volcano map (Fig 2A). For up-regulated DEGs, the GO terms for biological processes (BP) revealed that these genes were principally related to protein autophosphorylation, peptidyl-serine/tyrosine phosphorylation and peptidyl-serine/tyrosine modification. The enriched GO terms for cellular components (CC) included the chromosomal region, condensed chromosome and spindle. Meanwhile, the GO terms for molecular function (MF) enrichment showed that up-regulated DEGs were primarily related to protein serine/threonine kinase and tyrosine kinase activity (Fig 2B). For down-regulated DEGs, BP analysis showed

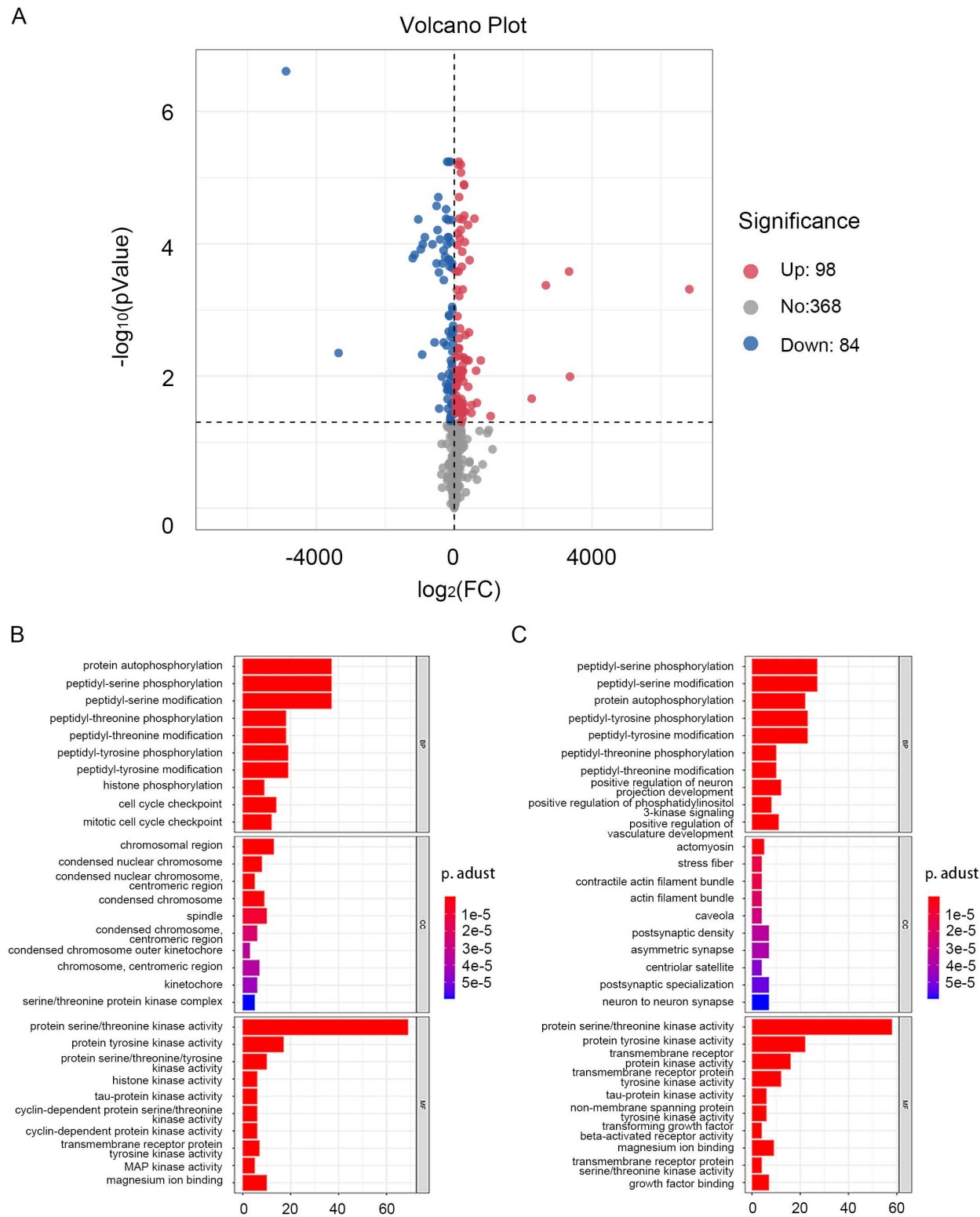

**Fig 2. Volcanic maps of DEGs and enrichment analysis result. (A)** Volcano plot of the DEGs. The blue dots represent downregulated DEGs. The red dots represent upregulated DEGs. **(B)** The results of upregulated DEGs in different functional groups. **(C)** The results of downregulated DEGs in different functional groups. DEGs, differentially expressed genes.

enrichment of peptidyl-serine/tyrosine phosphorylation and peptidyl-serine/tyrosine modification. CC analysis found that the DEGs were mainly involved in actomyosin, stress fiber and the contractile actin filament bundle, and MF analysis revealed that the DEGs were connected with protein serine/threonine kinase and protein tyrosine kinase activity (Fig 2C). Subsequently, the 182 DEGs were imported into STRING to filter the PPI complex and determine gene function at the protein level, along with the screened hub genes. The PPI network of DEGs consisted of 149 nodes and 633 edges. The top ten DEGs, including MTOR, CDK1, FYN, CHEK1, CDK2, CHEK2, ABL1, AURKA, AURKB, and PLK1, with a connective degree of ≥ 24 were identified as hub genes. The fold-changes of these ten core genes were respectively 79.78, 584.75, −308.59, 161.25, 167.15, 90.28, −338.23, 132.12, 444.21, 304.87. The relative expression matrix of these genes was visualized (Fig 3A).

Differential biological expression of the hub genes between MIBC and normal bladder tissue was verified by analyzing ten hub genes in the TCGA-based GEPIA database. The mRNA expression of seven hub genes, ABL1, AURKA, AURKB, CDK1, CHEK1, FYN, and PLK1, differed significantly between MIBC and normal bladder tissue (Fig 3B).

### 3.2 The landscape of immune infiltration in MIBC

The infiltration of 22 immune cell subpopulations was examined in MIBC tissue, and the difference between tumor and normal tissue was compared using the CIBERSORT algorithm. The proportion of each immune cell type varied significantly between groups (Fig 4A), and was weakly or strongly correlated with MIBC (Fig 4B). CD8 + T cells had a stronger positive correlation (Pearson correlation = 0.61) than activated memory CD4 + T cells and a stronger negative correlation ($p = −0.56$) than resting memory CD4 + T cells. CD8 + T cells also had a moderate negative correlation with M0 macrophages ($p = −0.48$). These results indicated that the immune response associated with MIBC is a complex cell network that proceeds in a closely regulated manner. Overall survival analysis of the IICs subsets showed that the level of CD8 + T cells ($p = 0.027$), activated memory CD4 + T cells ($p = 0.009$), regulatory T cells (Tregs) ($p = 0.015$) and M0 macrophages ($p = 0.005$) were significantly associated with patient survival. The Kaplan-Meier curve of significant immune cells is shown in Fig 4C.

### 3.3 Correlation between hub genes and IICs

The HPA database was used to determine the corresponding protein expression levels. Typical immunohistochemistry images for six proteins (CHEK1 was not founded in the HPA database) in MIBC and normal bladder samples were shown in Fig 5A. Four hub genes, ABL1, AURKA, CDK1, and FYN, had potential value as diagnostic markers for BCa. The correlation between four hub genes and IICs was investigated using the TIMER tool (Fig 5B). CD8 + T cell infiltration was positively correlated with the expression of ABL1, AURKA, CDK1 and FYN. Meanwhile, AURKA expression correlated positively with dendritic cell infiltration. ABL1 and FYN were significantly correlated with other immune infiltrating cells, including CD4 + T cells, macrophage, neutrophils, and dendritic cells.

### 3.4 AURKA identified as the potential target of Res by using qPCR and molecular docking

In order to further identify potential targets for the treatment of BCa, the expression levels of hub genes were measured using qPCR technology. The research results show that there are differences in the mRNA expression levels of four hub genes compared to normal bladder cells, especially AURKA and CDK1 are the most prominent (Fig 6A). The mechanism of action of CDK1 in BCa has been extensively and thoroughly reported [17–19]. Although several studies have shown that AURKA may be the target gene for BCa [20,21], its action mechanism has not been fully elucidated. Compared to CDK1, AURKA has greater research value and more potential for novel discoveries. Additional, it has been reported that Res inhibits the cell cycle process by targeting AURKA in breast cancer cells [22], which provides theoretical basis for how AURKA mediates the effect of Res in BCa and regulates the related signaling pathways. Therefore, AURKA was selected as the core target for the subsequent research.

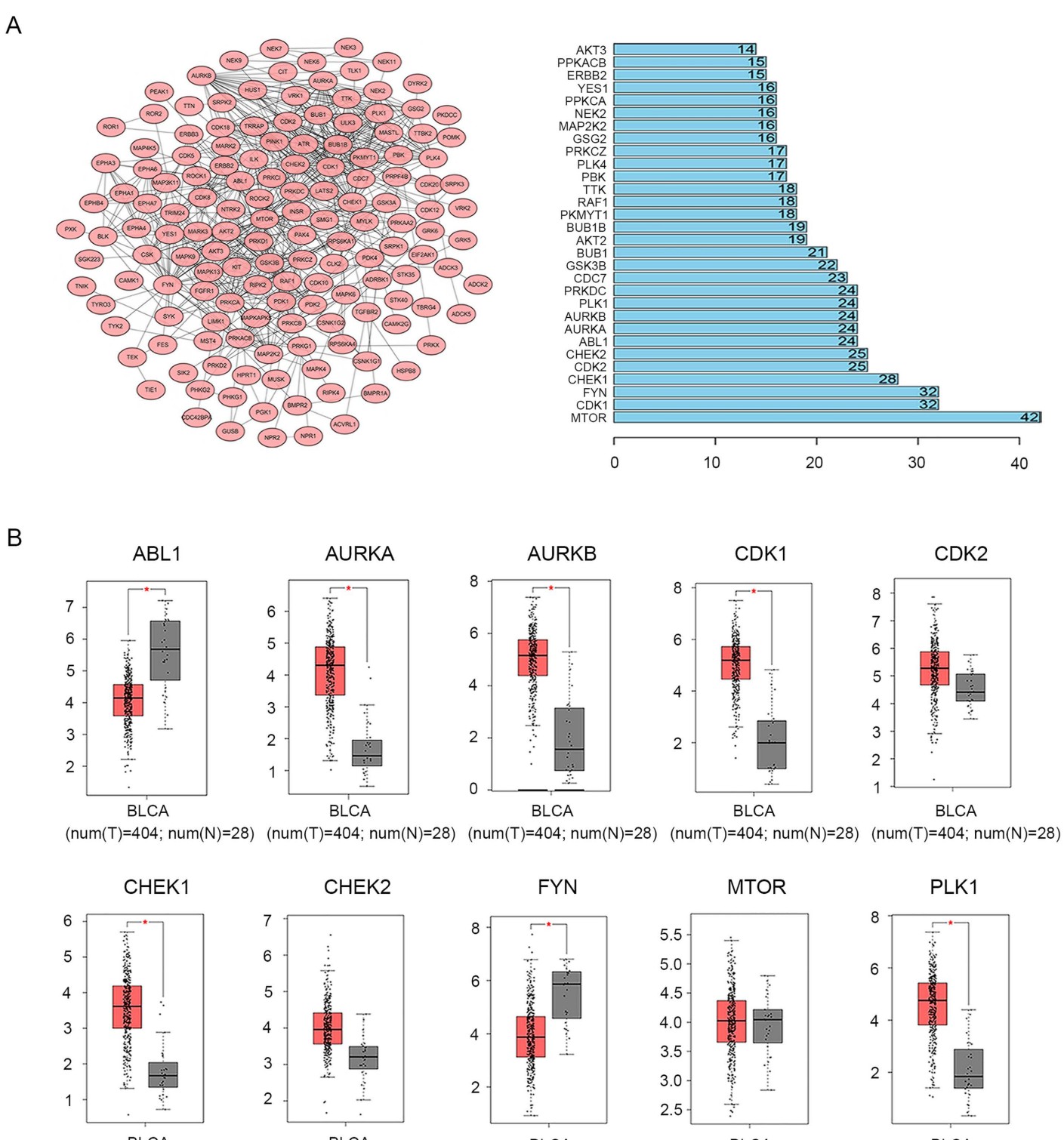

**Fig 3. Differential expression analysis of DEGs. (A)** Construction of PPI network and the top hub genes' relative expression matrix from GEO database. **(B)** Validation of the 10 hub genes by using the GEPIA-based TCGA database. The red color represents the tumor group and the grey color represents the normal group. DEGs, differentially expressed genes; PPI, Protein-protein Interaction; GEO, Gene Expression Omnibus. GEPIA, Gene Expression Profiling Interactive Analysis; TCGA, The Cancer Genome Atlas.

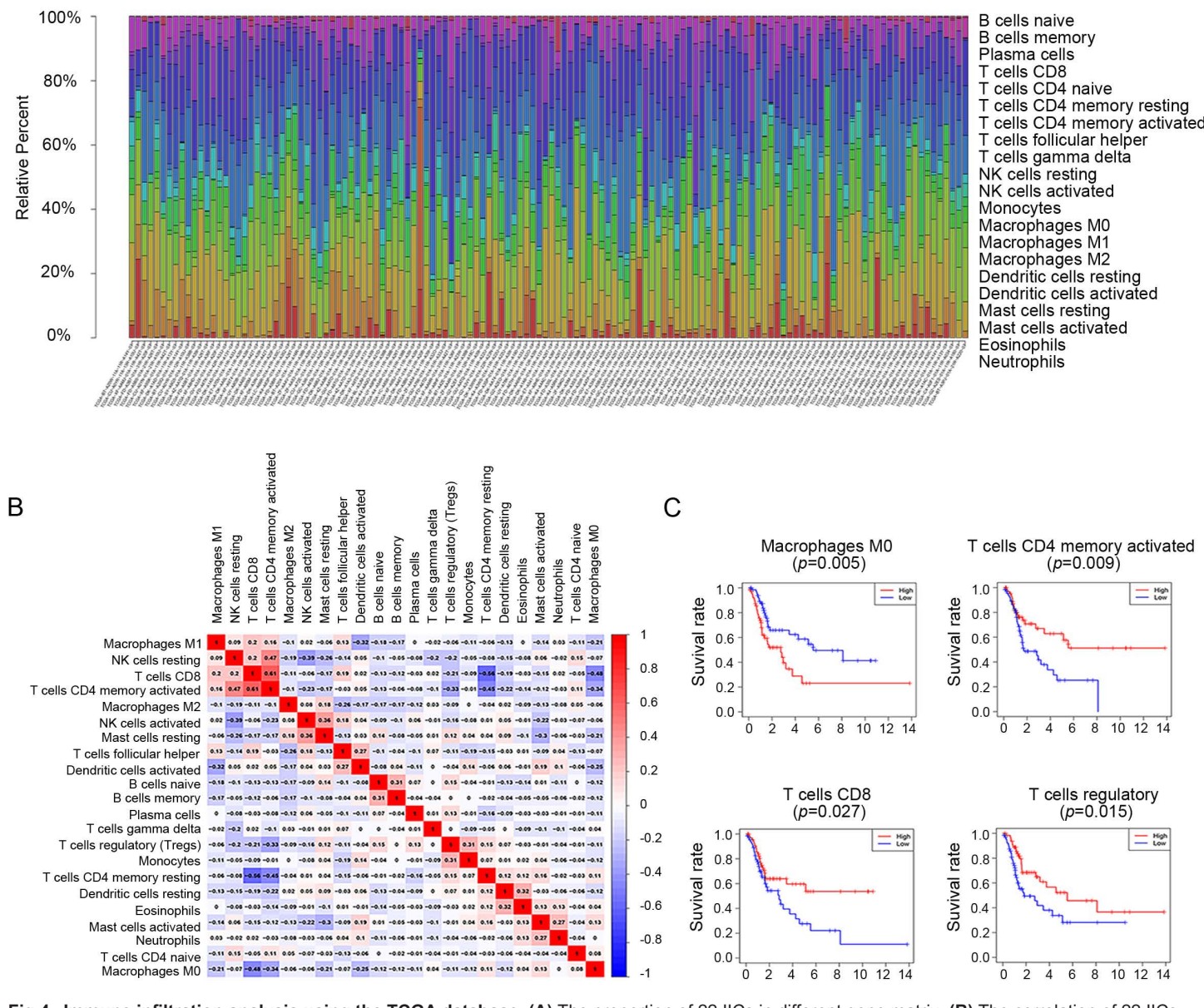

**Fig 4. Immune infiltration analysis using the TCGA database. (A)** The proportion of 22 IICs in different gene matrix. **(B)** The correlation of 22 IICs. **(C)** Identifi-cation of clinical implications of tumor IICs subsets by the overall survival analysis. TCGA, The Cancer Genome Atlas; IICs, Immune Infiltra-tion Cells.

The lower the free energy of substrate binding, the better the effect of binding. To determine the binding ability of Res with AURKA, this study used discovery studio to analyze the interaction forces of the Res and AURKA complex model. The results revealed that the molecular docking analysis identified 16 key interacting residues between Res and AURKA, which include critical residues for catalytic activity and allosteric regulation. Notably, the catalytic core residue Asp274 forms both hydrogen bonds and hydrophobic interactions with Resveratrol, acting as a 'dual-lock' to stabilize the confor-mation of the activation loop. Furthermore, Gly145 forms a backbone hydrogen bond with Res, inducing a steric effect that

**A**

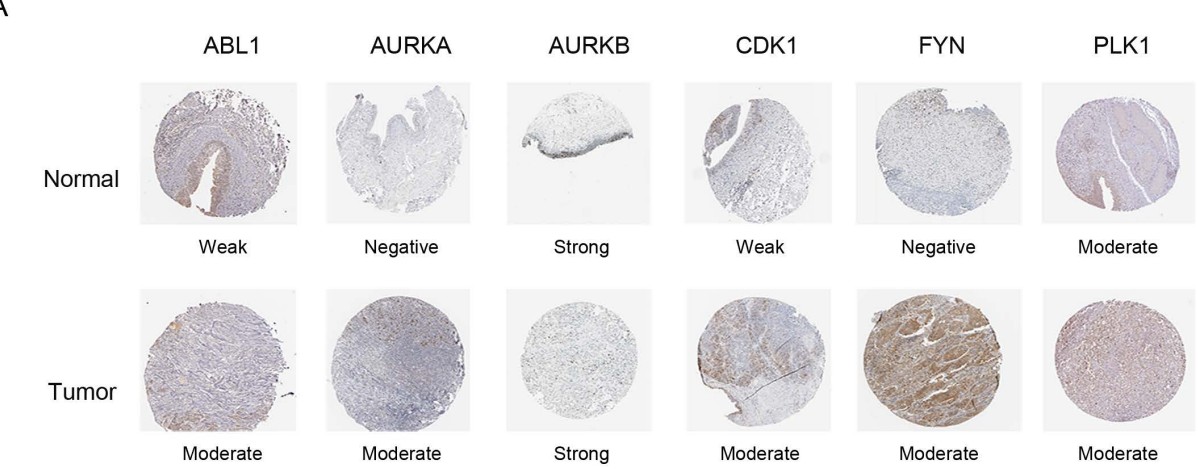

**B**

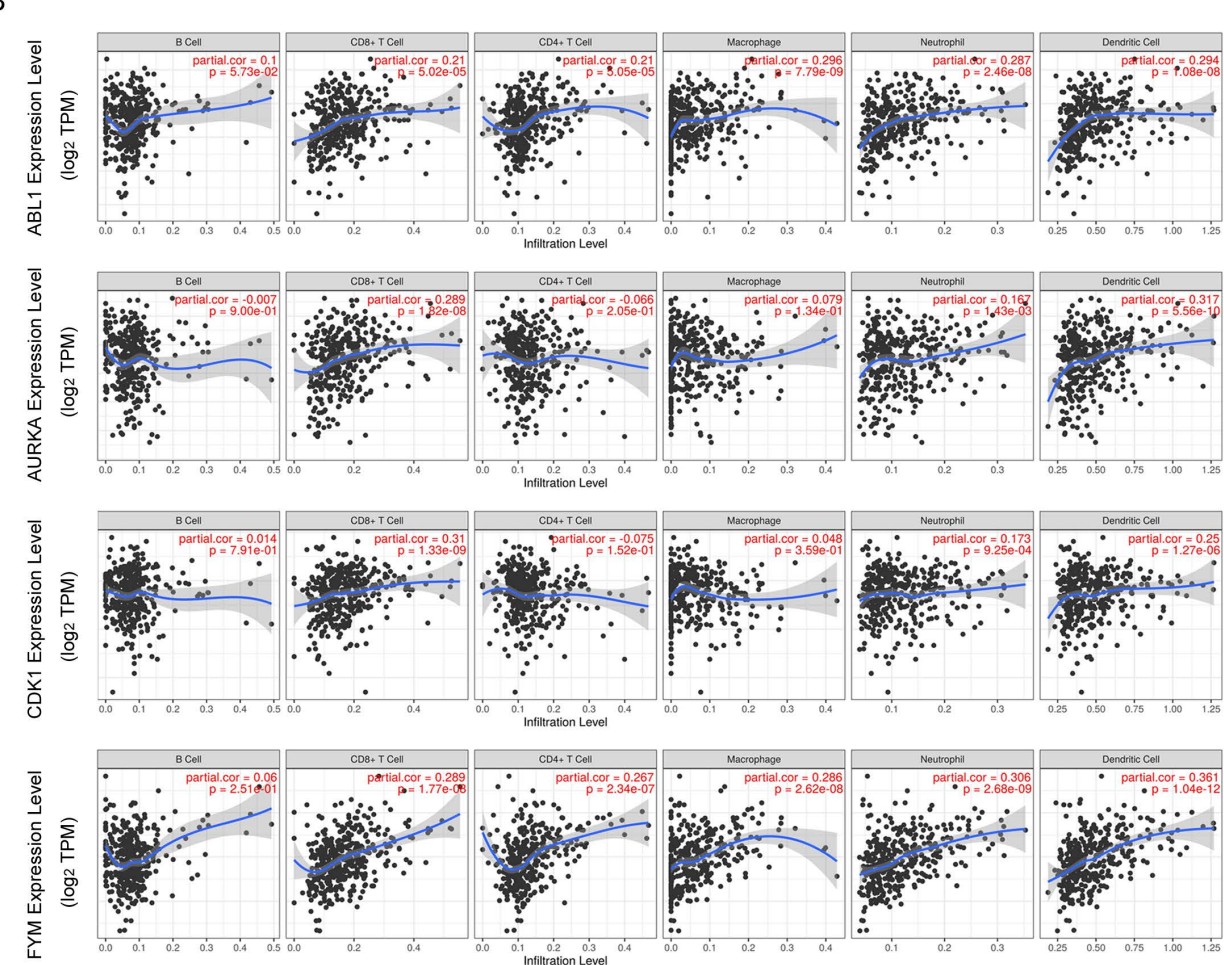

**Fig 5. Protein expression of DGEs and correlation between DGEs and TIICs. (A)** The protein expression level results of six hub genes by the Human Protein Atlas database. **(B)** Correlation between four DGEs and TIICs in bladder cancer. DEGs, differentially expressed genes; TIICs, Tumor Immune Infiltration Cells.

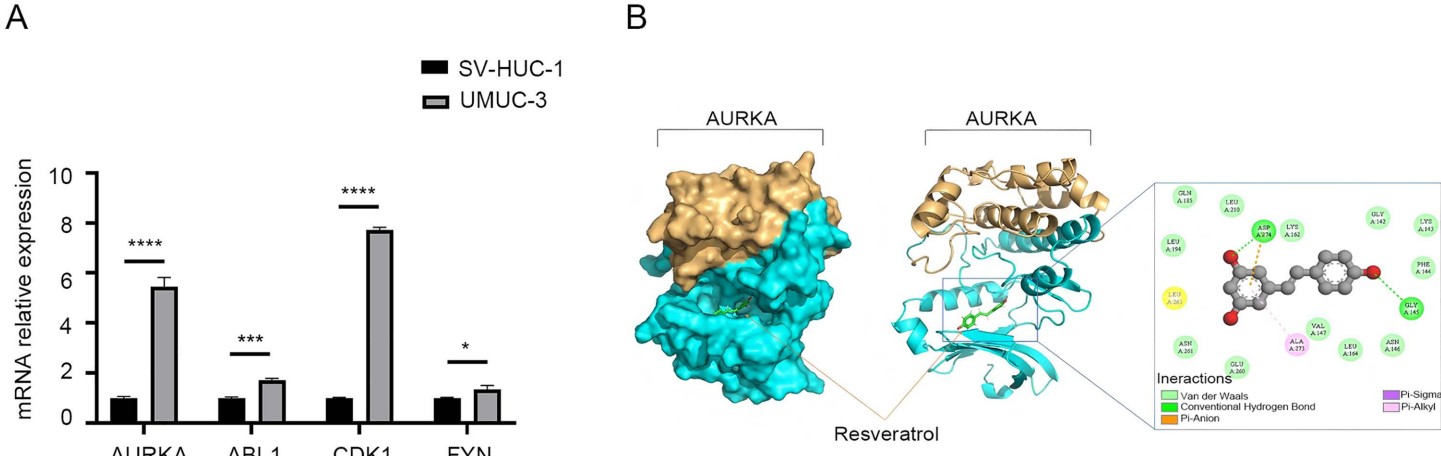

**Fig 6. qPCR analysis and computer molecular simulation docking. (A)** The differences in the mRNA relative expression levels of four hub genes compared to normal bladder cells by PCR. **(B)** The results of computer molecular simulation docking of Res and AURKA. SV-HUC-1, normal bladder epithelial cell; UMUC-3, bladder cancer cell.

promotes a contraction of the binding pocket and physically blocks the ATP entry channel. The residue Ala273 contributes to the binding stability by forming a stable hydrophobic stacking with Res. The synergistic effects of these multiple hydrogen bonds and hydrophobic forces construct a stable regulatory conformation, significantly enhancing the affinity of AURKA for Res and playing a crucial role in maintaining the stability of this complex (Fig 6B).

### 3.5  Res inhibits BCa cells and their cell cycle in a time- and dose-dependent manner

By calculating that the concentration of IC30 after 2 hours of Res treatment was 460.6μM and the concentration of IC50 was 640.5μM (Fig 7A); The concentration of IC30 after 24 hours of Res treatment was 132.4μM, and the concentration of IC50 was 146.7μM (Fig 7B). With the increase of MLN8237 concentration, the cell survival rate gradually decreased, and the IC50 concentration of MLN8237 was 49.75μM (Fig 7C). Compared with the control group, the apoptosis rates of the Res IC30 group, IC50 group and high concentration group at 2 hours and 24 hours were significantly increased, and in a dose-dependent manner (Fig 8A). Meanwhile, the cell cycles of the IC30 group and IC50 group treated with Res for 24 hours were significantly affected. The proportions of the G0/G1 phase increased significantly in the IC30 group and IC50 group, the proportion of the S phase in the high-concentration group cells increased and the proportion of G0/G1 phase decreased significantly ($P<0.05$). However, there were no significant differences in the proportions of the G0/G1 phase, S phase, and G2/M phase among the groups treated with Res for 2 hours (Fig 8B).

### 3.6  Res inhibits the expression of AURKA and known target genes in STAT3 signaling pathway

The results of STAT3 immunofluorescence staining are shown in Fig 9. The cell nuclei stained with DAPI appear blue under ultraviolet excitation, while positive expression is indicated by the red fluorescence of the corresponding fluorophore (DAPI excitation wavelength 330–380 nm, emission wavelength 420 nm, emits blue light; CY3 excitation wavelength 510–560 nm, emission wavelength 590 nm, emits red light). Compared with the black control group, the fluorescence intensity of STAT3 in both the IC50 Res group and the IC50 MLN8237 group decreased significantly, indicating that Res and AURKA inhibitor reduced the expression of STAT3 in cells.

The qPCR detection results are shown in Fig 10A. Compared with the control group, the IC30 Res group only had differences in the mRNA expression of CyclinD1 and AURKA. With the increase of Res concentration, the IC50 Res group

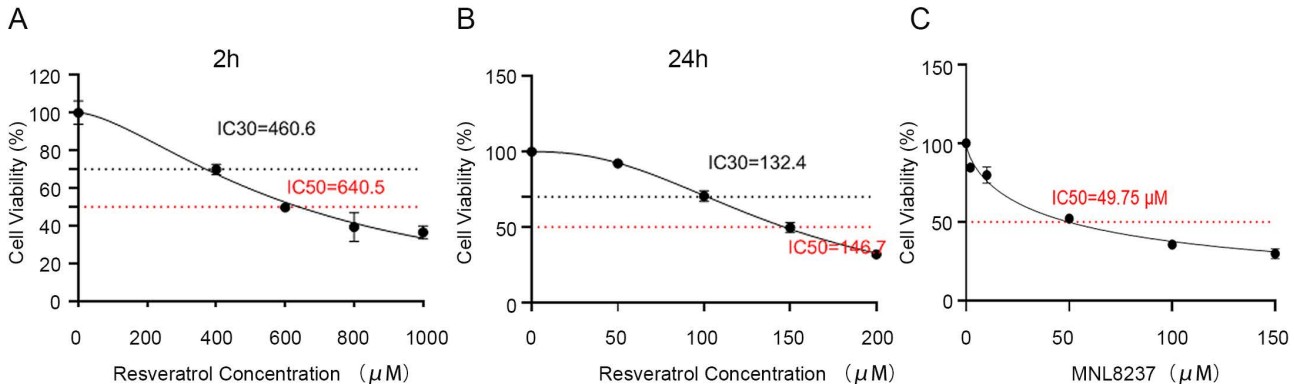

**Fig 7. Concentration detection of resveratrol and MLN8237. (A)** The concentration of IC30 and IC50 after 2 hours of Res treatment. **(B)** The concentration of IC30 and IC50 after 24 hours of Res treatment. **(C)** The IC50 concentration of MLN8237. IC30, 30% inhibitory concentration; IC50, Half maximal inhibitory concentration; MLN8237, AURKA inhibitor.

showed significant differences in the mRNA expression levels of AURKA and three downstream genes. The MNL8237 inhibitor group similarly could reduce the mRNA expression levels of the three genes, and there were significant differences in Cyclin D1 and Bcl-2. Interestingly, the result showed that MLN8237 can increase the mRNA expression level of AURKA, which may be related to the cell's feedback regulation to compensate for the transcription of AURKA.

The WB detection results are shown in Fig 10B. The molecular weights of AURKA, Bcl-2, CyclinD1, VEGF and GAPDH are 46KD, 26KD, 34KD, 35KD and 36KD respectively. Compared with the control group, the protein expression levels of AURKA in the three experimental groups decreased significantly. Meanwhile, the protein expression levels of three downstream genes (cyclinD1, VEGF and BCL-2) related to the STAT3 pathway also decreased significantly. This indicates that Res can induce tumor cell apoptosis by down-regulating AURKA and reducing STAT3 levels, and AURKA may be related to the STAT3 pathway.

## 4 Disscussion

At present, the treatment of MIBC remains a formidable clinical challenge, with high invasiveness leading to a persistently high recurrence and metastasis rate [23]. Relevant studies have shown that the 5-year survival rate of metastatic patients is lower than 15%, which urges us to urgently develop novel and cost-effective therapeutic strategies [24,25]. However, it should be noted that although resveratrol first demonstrated anti-metastatic potential as early as 1997, the related research progress remains limited so far. This underscores the critical need to identify its key direct molecular targets and the associated downstream signaling pathways in its anti-tumor effects [26,27]. Meanwhile, AURKA is considered to have valid biological function and prognostic significance in neuroblastoma, medullary thyroid carcinoma, hepatocellular carcinoma and other tumors [28–31]. As an important serine/threonine kinase, AURKA plays a critical role in the G2/M transition, promoting mitosis and other functions [32]. Although several previous studies have shown that AURKA may be a target gene of BCa, there are relatively few studies on how it affects the progression of BCa cells between natural anti-tumor drugs and related pathways [20,33]. In this study, a new therapeutic axis in BCa were elucidated through combing bioinformatics analysis with multi-level experimental verification: Our research not only calculated the amino acid sites where Res combined with AURKA, but also indicated that resveratrol could suppress the proliferation of BCa cells by down-regulating the expression of AURKA. We also discovered that resveratrol may exert its powerful anti-tumor effect by inhibiting the AURKA/STAT3 signaling pathway, thereby inhibiting the proliferation and cell cycle progression of bladder cancer cells. These findings not only provide mechanistic insights into the anti-tumor properties of resveratrol, but also highlight the importance of the AURKA/STAT3 axis as a promising therapeutic target for BCa.

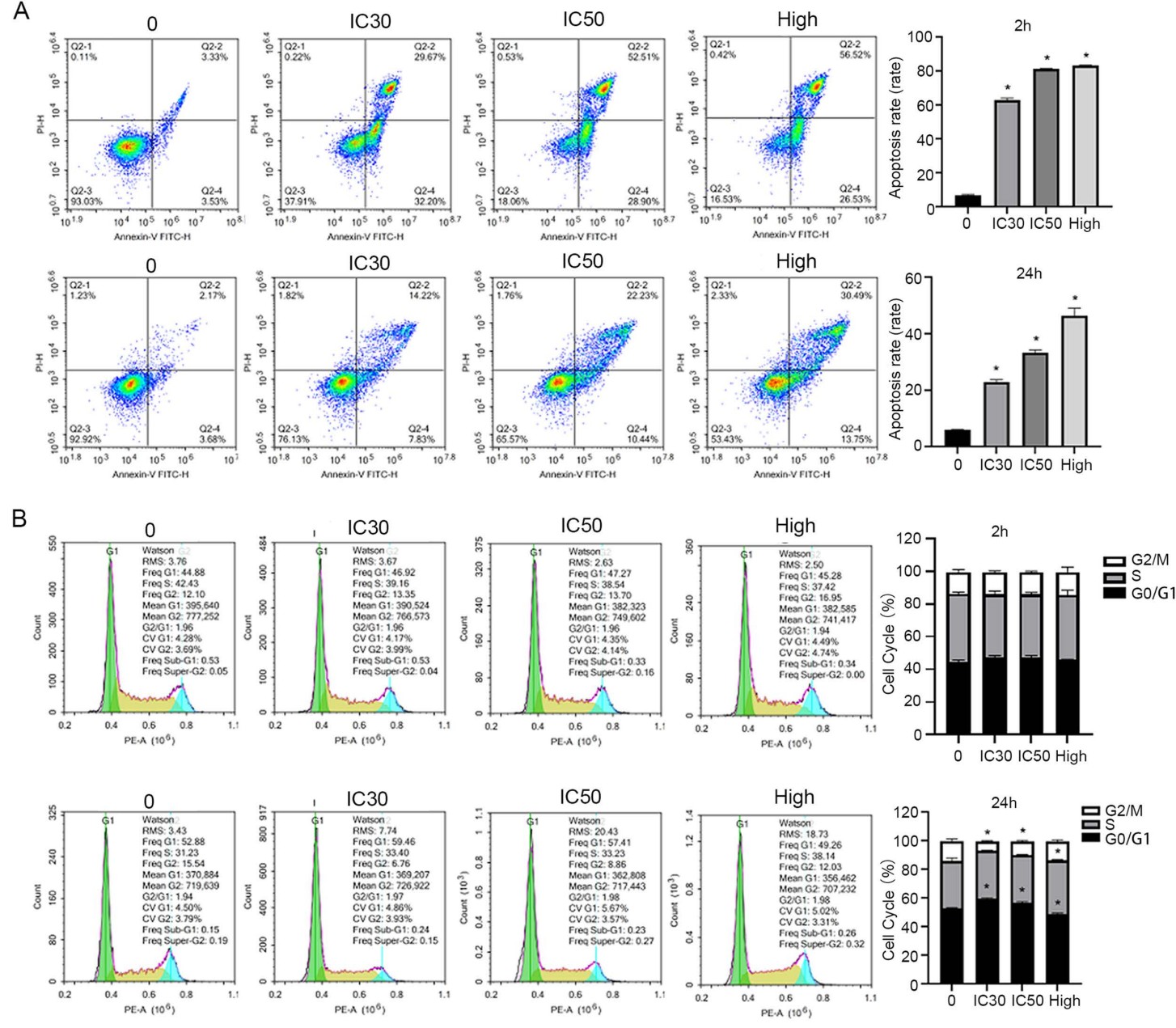

**Fig 8. Cell apoptosis and cell cycles detection. (A)** The results of cell apoptosis in four groups during 2 and 24 hours. **(B)** The results of cell cycles in four groups during 2 and 24 hours. Asterisks indicate significant differences from the control strain (*P < 0.05, Student's t test). G1/G0, Gap 1/Gap 0 phase; S, Synthesis phase; G2/M, Gap 2/M phase. IC30, 30% inhibitory concentration; IC50, Half maximal inhibitory concentration.

As an important tumor suppressor pathway, the STAT3 pathway has been widely confirmed that it plays a vital role in various common tumors including breast cancer, thyroid cancer, and cervical cancer through inhibiting cancer cell proliferation, metastasis, associating chemotherapy resistance and other aspects [34–36]. VEGF, Bcl-2 and Cyclin D1 have been confirmed to be the target genes which participate in the regulation of the STAT3 pathway [37,38]. VEGF is involved in the formation of tumor blood vessels in cancer tissues and is an important driver and target in the tumor

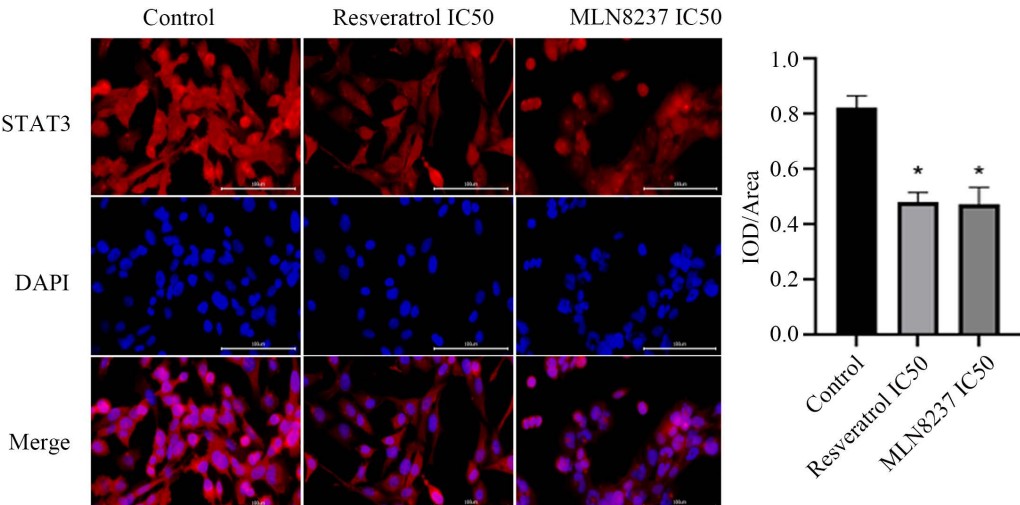

**Fig 9. Immunofluorescence staining.** The results of STAT3 immunofluorescence staining in three groups. The cell nuclei stained with DAPI appear blue under ultraviolet excitation, while positive expression is indicated by the red fluorescence of the corresponding fluorophore. Asterisks indicate significant differences from the control strain (*$P<0.05$, Student's *t* test). STAT3, Signal Transducer and Activator of Transcription 3.

vascular system [39]. Bcl-2 plays a key role in regulating the homeostasis between autophagy and apoptosis of cells. The overexpression of Bcl-2 is regarded as an independent adverse prognostic factor [40,41]. Cyclin D1 is a key regulatory factor for the transition from the G1 phase to the S phase in the cell cycle process [42]. Studies have shown that the continuous activation of STAT3 can promote the transformation from bladder carcinoma in situ to invasive BCa [37]. The STAT3 signaling pathway can also be involved in the formation of tumor inflammatory microenvironment, which is closely related to tumor progression and malignant transformation [43]. At the same time, Jiao Wu et al. found that Res can control the progression of thyroid cancer by regulating the expression of STAT3 [35]. Additionally, the AURKA inhibitor MLN8237 has been confirmed to inhibit the growth of breast cancer cells by altering the tumor microenvironment and suppressing the AURKA kinase. This is unique as the only agent currently in Phase III clinical trials [44]. Based on above theoretical foundation, the GEO and TCGA large databases were used to analyze and screen out the differentially expressed gene AURKA in bladder cancer from multiple perspectives. Experimental validation via qPCR further demonstrated that AURKA mRNA expression was markedly higher in the UMUC-3 cell line than in the normal SV-HUC-1 cell line. These results underscore the significance of AURKA and justify its selection for further in-depth analysis.

Subsequently, the IC50 concentrations of Res and the AURKA inhibitor MLN8237 were determined. Molecular docking simulations confirmed that Res binds to AURKA at multiple sites, establishing a fundamental basis for the subsequent experiments. Meanwhile, our results demonstrated that Res significantly suppressed the mRNA expression of AURKA, thereby influencing the cell cycle and inducing apoptosis. Furthermore, both Res and MLN8237 were found to effectively inhibit the STAT3 signaling pathway, as evidenced by a significant reduction in STAT3 fluorescence intensity and the downregulation of its downstream target proteins, including VEGF, Bcl-2, and Cyclin D1. Previous studies have shown that AURKA, as a factor regulating the cell cycle and apoptosis, has been found to participate in the drug resistance of breast cancer together with the STAT3 pathway [45]. Building upon this, our study is to link this regulatory network to the clinical challenge of targeted therapy resistance in bladder cancer, thereby proposing a novel scientific hypothesis: AURKA and the STAT3 pathway may act synergistically to mediate this resistance. Consequently, this work not only provides a crucial theoretical foundation and a new direction for future validation of this hypothesis but also offers strong data to support the

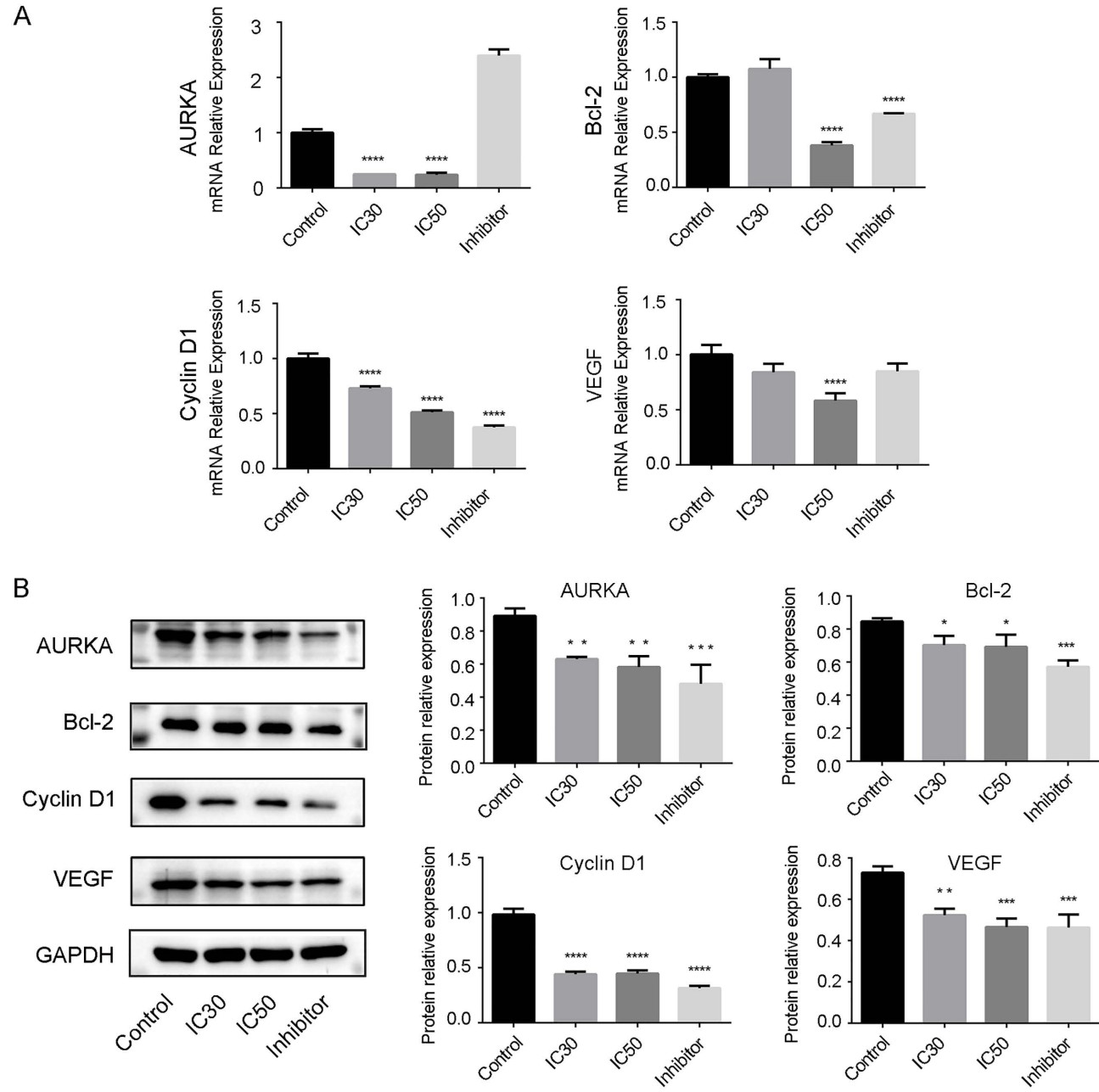

**Fig 10. Detection of AURKA and STAT3 pathway genes by qPCR and western blot. (A)** The qPCR detection results of AURKA, VEGF, Cyclin D1 and Bcl-2 in four groups. **(B)** The Western Blot detection results of AURKA and three STAT3 downstream genes (cyclinD1, VEGF and BCL-2) in four groups. Asterisks indicate significant differences from the control strain (*$P<0.05$; **$P<0.01$; ***$P<0.001$; ****$P<0.0001$, Student's $t$ test). STAT3, Signal Transducer and Activator of Transcription 3.

potential clinical therapeutic strategies, such as monotherapy with Res or combination therapies targeting the AURKA/STAT3 axis, to improve treatment outcomes for bladder cancer patients.

It is undeniable that there are still some deficiencies in our research. The latest research on the STAT3 pathway has revealed that in addition to the typical tyrosine phosphorylation activation pathway (P-STAT3), there is also a

non-phosphorylation pathway (U-STAT3) activated by multiple target genes [46]. Although our study indicates that Res inhibits the AURKA/STAT3 axis in bladder cancer cells, the exact molecular mechanism of STAT3 activation mediated by AURKA still needs to be further elucidated. At the same time, our current experimental verification is limited to a single bladder cancer cell line (UMUC-3). Future researches should expand to multiple cell lines representing different molecular subtypes and verify the universality of our findings by constructing overexpressed or low-expressed cells. Moreover, Res currently has many defects such as poor stability, low bioavailability and so on [47]. However, along with the continuous development of various encapsulation methods, it will provide broad research prospects for subsequent in vivo experiments and clinical studies through preparing stable and efficient Res nanodelivery systems [48,49].

## 5 Conclusion

In conclusion, this study provides compelling evidence that the anti-tumor effects of Res in bladder cancer are mediated through targeting of AURKA and subsequent suppression of the AURKA/STAT3 signaling axis. This work deciphers a key mechanism of action for a promising natural compound and these results will provide a theoretical foundation for the structural optimization of Res and the development of multi-target drugs for clinical application.

## Supporting information

**S1 Fig. S1_raw_images - Repeat 1: The original images of Fig 10B, repeat 1.**
(TIF)

**S2 Fig. S1_raw_images - Repeat 2: The original images of Fig 10B, repeat 2.**
(TIF)

**S3 Fig. S1_raw_images - Repeat 3: The original images of Fig 10B, repeat 3.**
(TIF)

**S4 Fig. S2_raw_images - Repeat 1: The original images of Fig 9, repeat 1.**
(TIF)

**S5 Fig. S2_raw_images - Repeat 2: The original images of Fig 9, repeat 2.**
(TIF)

**S6 Fig. S2_raw_images - Repeat 3: The original images of Fig 9, repeat 3.**
(TIF)

**S7 Fig. S3_raw_images - Repeat 1: The original images of Fig 8, repeat 1.**
(TIF)

**S8 Fig. S3_raw_images - Repeat 2: The original images of Fig 8, repeat 2.**
(TIF)

**S9 Fig. S3_raw_images - Repeat 3: The original images of Fig 8, repeat 3.**
(TIF)

**S1 Appendix. The raw data of 182 differentially expressed genes.**
(XLSX)

**S2 Appendix. The CIBERSORT results of TCGA database.**
(XLSX)

**S3 Appendix. Fig 6A: The raw data of Fig 6A.**
(XLSX)

**S4 Appendix. Fig 7: The raw data of Fig 7.**
(XLSX)

**S5 Appendix. The raw data of Figs 8 and 9.**
(XLSX)

**S6 Appendix. Fig 10A: The raw data of Fig 10A.**
(XLSX)

**S7 Appendix. Fig 10B: The raw data of Fig 10B.**
(XLSX)

**S1 File. PLOSOne human subjects research checklist.**
(DOCX)

## Author contributions

**Conceptualization:** Chao Feng, Ligang Ren.

**Data curation:** Chao Feng, Guodong Chen.

**Formal analysis:** Chao Feng, Yan Shu.

**Funding acquisition:** Chao Feng.

**Investigation:** Jing Chen.

**Methodology:** Wenxin Ye.

**Project administration:** Chao Feng, Ligang Ren.

**Software:** Chao Feng.

**Validation:** Chao Feng, Ligang Ren.

**Writing – original draft:** Chao Feng.

**Writing – review & editing:** Ligang Ren.

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
