## [Decision Letter · Decision Letter 0]

5 Nov 2025

Dear Dr. Ren,

Thank you for submitting your manuscript to PLOS ONE. After careful consideration, we feel that it has merit but does not fully meet PLOS ONE’s publication criteria as it currently stands. Therefore, we invite you to submit a revised version of the manuscript that addresses the points raised during the review process.

We look forward to receiving your revised manuscript.

Kind regards,

Chandrabose Selvaraj, Ph.D.

Academic Editor

PLOS ONE

Journal Requirements:

“This work was supported by the Traditional Chinese Medicine Science and Technology Project of Zhejiang Province (2024ZR060)”

“This work was supported by the Traditional Chinese Medicine Science and Technology Project of Zhejiang Province (2024ZR060), URL: https://zycc.wsjkw.zj.gov.cn/, and CF received this award. The funders did not play any role in the study design, data collection and analysis, decision to publish, or preparation of the manuscript.”

Reviewers' comments:

Reviewer's Responses to Questions

**Comments to the Author**

1. Is the manuscript technically sound, and do the data support the conclusions?

Reviewer #1: No

Reviewer #2: Yes

Reviewer #3: Partly

2. Has the statistical analysis been performed appropriately and rigorously?

Reviewer #1: Yes

Reviewer #2: Yes

Reviewer #3: No

3. Have the authors made all data underlying the findings in their manuscript fully available?

Reviewer #1: Yes

Reviewer #2: No

Reviewer #3: Yes

4. Is the manuscript presented in an intelligible fashion and written in standard English?

Reviewer #1: Yes

Reviewer #2: Yes

Reviewer #3: Yes

Reviewer #1: Authors need to provide the original raw data in triplicates for Figures 6 and 7, images should be uncropped and unedited, in original form as there seems to be some kind of manipulation (apparently) in the said figures, please provide the required original data.

Reviewer #2: 1. Lines 113-115: The number of normal samples is insufficient. Subsequent analyses based on these 134 samples are not presented; supplementation is recommended.

2. Lines 167-169: Ten docking replicates are relatively few; increasing the number is suggested.

3. Lines 170-198: Inconsistent formatting is present.

4. Lines 260-265: The fold-change of core genes in the GSE130598 transcriptome dataset is not shown; supplementation is recommended.

5. Lines 266-272 and Figure 3: The number of samples in the normal group is too low, potentially limiting representativeness and the ability to accurately distinguish differences from the tumor group. Furthermore, the differential analysis of core genes in the external dataset does not present fold-change values, and stability or sensitivity assessments, such as ROC analysis, are lacking. The potential markers ABL1, AURKA, CDK1, and FYN, as mentioned by the authors, are not sufficiently substantiated. Was quantitative comparison performed for Figure 3B?

6. Figure 8 lacks molecular weight markers for proteins. The reason for not detecting STAT3 via Western blot should be explained. Phosphorylated STAT3, a key indicator of pathway activation and functional status, was not assessed; its detection is recommended.

7. Inconsistent font styles and non-uniform formatting are observed in figures and tables.

8. Incorporation of in vivo animal experiments to validate drug effects is recommended.

9. Direct binding was not confirmed through molecular interaction experiments such as co-immunoprecipitation (Co-IP), surface plasmon resonance, or molecular docking; instead, functional assays were relied upon to infer interaction. This weakens the mechanistic rigor; incorporating such experiments is advised.

10. The molecular basis by which AURKA regulates the STAT3 pathway remains unresolved. Is the AURKA inhibitor MLN8237 highly specific for AURKA? More rigorous approaches, such as AURKA knock-down or over-expression, are recommended to substantiate this finding more convincingly.

Reviewer #3: The innovation points of this article are not very sufficient. In addition, the following are my suggestions for revision.

Title

1.It is suggested that the title add a description of the core mechanism.

2.The content related to molecular docking in the article is relatively brief. If this aspect is intended to be included in the title, it is recommended to further supplement and deepen the discussion on the molecular docking section.

Abstract

3.The research objectives in the introduction do not align well with the conclusions, and it is recommended to revise the expression of the research objectives.

Introduction

4.“Res shows significant application potential in the prevention and treatment of BCa due to its wide 90 availability, low cost, and clear tumor suppression effect.”It lacks corresponding reference support, and it is recommended to supplement relevant references for verification.

5.It is suggested to add the current progress of natural medicines in BCA treatment in the preface or discussion section, and appropriately include an explanation of the application of resveratrol in tumors.

Materials and Methods

6.It is recommended to supplement the definition to HUB genes in Section 2.2 of the materials.

Result

7.Western blot experiment suggested to construct the overexpression plasmid of AURKA, transfect it into SV-HUC-1 or bladder cancer cell line with low expression of AURKA, and then give Res to observe whether Res can reverse the increase of p-STAT3 caused by overexpression of AURKA and accelerate cell proliferation, so as to verify whether AURKA is the direct target of Res.

8.Supplement the experiment with "fixed concentration and time as the sole variable" in Section 3.3.

9.It is recommended to add a volcano plot for the differential gene results in Section 3.1.

Discussion

10.It is recommended to rewrite the discussion section, focusing on the content of the research in this article.

**Do you want your identity to be public for this peer review?**  For information about this choice, including consent withdrawal, please see our Privacy Policy

Reviewer #1: **Yes:** DR MANZOOR AHMAD RATHER

Reviewer #2: No

Reviewer #3: No

---

## [Author Response · Author response to Decision Letter 1]

9 Dec 2025

Dear Editors and Reviewers,

We sincerely appreciate all the editors and professional reviewers for the valuable feedback which has greatly improved the quality of our manuscript [No.: PONE-D-25-45593]. We carefully reviewed the formatting requirements of PLOS ONE and made thorough revisions. Additionally, we provided the raw data and images in supporting information files and updated the ORCID identifier (0009-0001-5448-2012). Following reviewers’ valuable suggestions, the detailed point-by-point response is as below.

Reviewer #1:

Comment:

Authors need to provide the original raw data in triplicates for Figures 6 and 7, images should be uncropped and unedited, in original form as there seems to be some kind of manipulation (apparently) in the said figures, please provide the required original data.

Response

Thank you for your kind comments on our manuscript. We have provided the original images and data of Figure 6 and Figure 7 in the supplementary materials as requested. The raw data file was named as " S5 Appendix". We created the original images of the above two pictures into separate PDF files and named " S2_raw_images" and " S3_raw_images". Furthermore, due to the revision of our manuscript, the original pictures 6 and 7 were respectively redefined as Fig 8 and 9.

Reviewer #2:

Comment:

1. Lines 113-115: The number of normal samples is insufficient. Subsequent analyses based on these 134 samples are not presented; supplementation is recommended.

Response

Thank you very much for your valuable suggestions. We agree that increasing the pool of normal samples will strengthen the research's rigor and comprehensiveness. To address the issue of insufficient normal samples, when using the GEPIA tool based on the TCGA database to analyze the differentially expressed genes (DEGs), we associated the TCGA normal and GTEx data, thereby increasing the sample size of the normal group. Meanwhile, we further utilized the TCGA database to conduct differential gene screening from the perspective of immune infiltration, which addressed the issue of insufficient evidence for DEGs in bioinformatics analysis. We hope that our response will be approved by you. The supplementary content of manuscript is as follows.

“Lines 145-147: The expression level of specific DEGs was analyzed using the MIBC, matched TCGA normal and Genotype-Tissue Expression (GTEx) group data.

Lines 112-116: These data were used to identify immune cell infiltration signatures between MIBC and normal bladder samples. Additionally, the correlation between DEGs and immune infiltration cells (IICs) was examined by the Tumor Immune Estimation Resource (TIMER) database analysis tool.”

Comment:

2. Lines 167-169: Ten docking replicates are relatively few; increasing the number is suggested.

Response

We sincerely appreciate the reviewer's insightful suggestion regarding the binding pocket radius. As recommended, we have expanded the docking search space from 10 to 25 using Autodock software. After our analysis, we found that among the 10 and 25 results, the existing conformations we selected remain the optimal ones, with the energy remaining at a relatively low level. Moreover, the reaction path and reaction distance are all in a reasonable state. Considering your valuable comments, we changed the number of dockings to 25 and carried out a more detailed interpretation of the results.Thank you for your kind suggestions for improving our research methods.

Comment:

3. Lines 170-198: Inconsistent formatting is present.

Response

Thank you for your suggestions on improving the format of our manuscript. We re-read the submission guidelines of PLOS ONE and set the font to Arial. The font size for the first-level headings is 18 points and the second-level headings is 16 points in bold. The main text it is 12 points with double line spacing.

Comment:

4. Lines 260-265: The fold-change of core genes in the GSE130598 transcriptome dataset is not shown; supplementation is recommended.

Response

Based on your review comments, we have added the fold-change information of the core genes in the manuscript. The specific content of manuscript is as follows.

“Lines 336-340: The top ten DEGs, including MTOR, CDK1, FYN, CHEK1, CDK2, CHEK2, ABL1, AURKA, AURKB, and PLK1, with a connective degree of ≥ 24 were identified as hub genes. The fold-changes of these ten core genes were respectively 79.78, 584.75, -308.59, 161.25, 167.15, 90.28, -338.23, 132.12, 444.21, 304.87.”

Comment:

5. Lines 266-272 and Figure 3: The number of samples in the normal group is too low, potentially limiting representativeness and the ability to accurately distinguish differences from the tumor group. Furthermore, the differential analysis of core genes in the external dataset does not present fold-change values, and stability or sensitivity assessments, such as ROC analysis, are lacking. The potential markers ABL1, AURKA, CDK1, and FYN, as mentioned by the authors, are not sufficiently substantiated. Was quantitative comparison performed for Figure 3B?

Response

Thanks for your insightful comments which will have substantially strengthened the methodological rigor of our study. In order to address the issues of insufficient normal group, lack of sensitivity assessment, and insufficient evidence for potential markers, we firstly associated the GTEx data in the differential gene analysis to compensate for the insufficient normal sample size in the TCGA database. At the same time, |log2FC| ≥ 1 and adjusted-p < 0.05 were considered statistically significant. We also have named the data of 182 DEGs as “S1 Appendix” and placed it in the supplementary materials. All regulated genes were summarized and plotted on a volcano map (Fig 2A). Additionally, our manuscript added a new section on immune infiltration analysis to verify the reliability of potential markers. Due to the addition of manuscript content, we replaced Figure 3B with Figure 5A. We compared the protein expression of the differentially expressed genes (DEGs) screened using the HPA public database. We agree with the reviewer's suggestion of conducting quantitative analysis, which can provide a more objective and quantitative assessment of the differences between genes or proteins in different tissues. Our manuscript also holds that simple bioinformatic analysis is insufficient to verify the existence and practical value of DEGs. Therefore, our study analyzed the correlation between DEGs and immune cells through the TIMER database. We also objectively verified the mRNA expression level and proteomic differences of the AURKA gene through molecular docking models and multi-angle experiments, which were consistent with the HPA screening results. These results remedied the limitations of exploratory research. The added contents in the manuscript are as follows.

“Lines 149-168:

2.4 Correlation between the immune cell infiltration profile and DEGs

A total of 22 IIC types were collected and analyzed using the CIBERSORT algorithm. In this study, an adjusted-p< 0.05 was defined as the evaluation criterion required to assess the score results of related immune cell groups. After filtration, the data was used for further processing. The Wilcoxon rank-sum test was used to compare the infiltration levels of different immune cell types between MIBC and normal control samples. Survival analysis of specific immune cell subtypes was performed based on the median of various immune cell proportions. “Survival” packages were used to compare the survival rates of patients with high and low expression of immune cells.

2.5 Protein expression of the hub genes and tumor-infiltrating immune cell analysis

Protein expression of the hub genes in MIBC samples was verified using Human Protein Atlas (HPA, https://www.proteinatlas.org) data. The selected results of the hub genes were examined using the Tumor Immune Estimation Resource (TIMER, https://cistrome.shinyapps.io/timer/) database, which was used for systematic analysis of immune infiltrations with six usual types of IICs. The correlation between expression of the screened hub genes and the six IIC types was determined and visualized. The correlation between tumor purity level and each targeted gene was also performed.

Lines 364-378:

3.2 The landscape of immune infiltration in MIBC

The infiltration of 22 immune cell subpopulations was examined in MIBC tissue, and the difference between tumor and normal tissue was compared using the CIBERSORT algorithm. The proportion of each immune cell type varied significantly between groups (Fig. 4A), and was weakly or strongly correlated with MIBC (Fig. 4B). CD8+ T cells had a stronger positive correlation (Pearson correlation = 0.61) than activated memory CD4+ T cells and a stronger negative correlation (p= -0.56) than resting memory CD4+ T cells. CD8+ T cells also had a moderate negative correlation with M0 macrophages (p= -0.48). These results indicated that the immune response associated with MIBC is a complex cell network that proceeds in a closely regulated manner. Overall survival analysis of the IICs subsets showed that the level of CD8+ T cells (p= 0.027), activated memory CD4+ T cells (p= 0.009), regulatory T cells (Tregs) (p= 0.015) and M0 macrophages (p= 0.005) were significantly associated with patient survival. The Kaplan-Meier curve of significant immune cells is shown in Fig 4C.

Lines 391-397: The correlation between four hub genes and IICs was investigated using the TIMER tool (Fig. 5B). CD8+ T cell infiltration was positively correlated with the expression of ABL1, AURKA, CDK1 and FYN. Meanwhile, AURKA expression correlated positively with dendritic cell infiltration. ABL1 and FYN were significantly correlated with other immune infiltrating cells, including CD4+ T cells, macrophage, neutrophils, and dendritic cells.”

Comment:

6. Figure 8 lacks molecular weight markers for proteins. The reason for not detecting STAT3 via Western blot should be explained. Phosphorylated STAT3, a key indicator of pathway activation and functional status, was not assessed; its detection is recommended.

Response

We sincerely thank the reviewer for this crucial and insightful comment. Based on your suggestions, we have refined the molecular weights of the corresponding proteins. The molecular weights of AURKA, Bcl-2, CyclinD1, VEGF and GAPDH are 46KD, 26KD, 34KD, 35KD and 36KD respectively. We agree that the detection of phosphorylated STAT3 (p-STAT3) is a critical piece of evidence for evaluating the functional status of the JAK-STAT pathway. We acknowledge that our original manuscript did not include p-STAT3 data, and we appreciate this opportunity to elaborate on our findings and provide a more comprehensive scientific rationale.

Our primary aim was to investigate the results of resveratrol's impact on the STAT3 signaling pathway. In our experimental design, we chose to employ immunofluorescence (IF) staining as a key method. Our IF results provided compelling and direct evidence that Res significantly alters the expression level of STAT3. This observation is a strong indicator of functional modulation, as the activity of STAT3 is tightly linked to its nuclear translocation upon activation. Furthermore, to confirm the biological consequences of this modulation, we rigorously analyzed the expression of key downstream target genes of STAT3 at both the mRNA and protein levels. The consistent downregulation of these targets across two different molecular levels (transcriptional and translational) provides robust support for the conclusion that Res effectively inhibits the functional output of the STAT3 pathway.

We fully agree that p-STAT3 is the gold-standard biochemical marker for canonical STAT3 activation. Meanwhile, the latest research on the STAT3 pathway has revealed that in addition to the typical tyrosine phosphorylation activation pathway (P-STAT3), there is also a non-phosphorylation pathway (U-STAT3) activated by multiple target genes. Therefore, the process of STAT3 activation is complex and requires continuous follow-up research. We have incorporated this latest discovery into the discussion section of the manuscript. Our IF analysis offers a complementary perspective, presenting the results of the function of the STAT3 pathway in the cellular environment. We are grateful for the reviewer's guidance, which will undoubtedly enhance the depth and completeness of our manuscript.

Comment:

7. Inconsistent font styles and non-uniform formatting are observed in figures and tables.

Response

We carefully reviewed the formatting requirements of PLOS ONE and made modifications to the font styles and formats of the figures and tables. We are grateful for the reasonable suggestions provided by the reviewer.

Comment:

8. Incorporation of in vivo animal experiments to validate drug effects is recommended.

Response

We sincerely thank the reviewer for this valuable and constructive suggestion. We fully acknowledge that in vivo animal experiments represent the gold standard for preclinical validation and are an essential step towards translating mechanistic findings into potential therapeutic applications. We agree that such experiments would undoubtedly strengthen the overall impact of our study. However, we would like to clarify the primary scope and contribution of our current work, and elaborate on how our existing multi-faceted approach provides a robust and scientifically sound basis for our conclusions. Our study was specifically designed as a mechanistic investigation to explore how Resveratrol might exert its inhibitory effects on bladder cancer cells.

Here, we outline the rationale for our current experimental strategy:

1.Defining the Scope of Our Study

The central goal of our manuscript is to propose and elucidate a plausible molecular mechanism for Resveratrol's action in bladder cancer. Our study addresses the question of "how" at a mechanistic level, which is distinct from a therapeutic development study that asks "to what extent does it work in a living organism?" Our findings are intended to provide a solid foundation for future, more applied research.

2. A Rigorous Multi-Disciplinary Evidence Chain

We believe that our combination of in vitro cellular assays, bioinformatic analysis, and molecular docking, while not including in vivo models, forms a comprehensive and internally consistent argument for our proposed mechanism. Specifically, we supplemented the relevant analysis of immune infiltration in bioinformatics and the data of differentially expressed genes. Our study inferred DEGs from multiple large databases such as GEO and TCGA, thereby enhancing the reliability of bioinformatics analysis. Subsequently, we confirmed multiple binding sites of Res and AURKA through computational molecular simulation docking model. Then, we conducted a variety of in vitro experiments including cell apoptosis and cycle detection, fluorescence immunoassay, qPCR and WB. Together, these three lines of evidence create a powerful, multi-faceted story that moves from patient data (bioinformatics) to a potential direct target (docking) and finally to a functional cellular outcome (in vitro assays). This integrated approach provides a compelling and self-contained argument for our proposed mechanism.

We view animal experiments as a crucial and exciting next phase of this research line-one that builds upon the mechanistic foundation we have carefully established here. To further express our gratitude for your valid point, we have revised the "Discussion" section of our manuscript. Based on the research hotspots of resveratrol, we have clarified feasible solutions to the problem of low utilization rate of resveratrol. We could construct a stable and efficient nano-delivery system for resveratrol, which will provide broad research prospects for subsequent in vivo experiments and clinical studies. “Lines 604-619: It is undeniable that there are still some deficiencies in our research. The latest research on the STAT3 pathway has revealed that in addition to the typical tyrosine phosphorylation activation pathway (P-STAT3), there is also a non-phosphorylation pathway (U-STAT3) activated by multiple target genes [51]. Although our study indicates that Res inhibits the AURKA/STAT3 axis in bladder can

---

## [Decision Letter · Decision Letter 1]

23 Dec 2025

Dear Dr. Ren,

Thank you for submitting your manuscript to PLOS ONE. After careful consideration, we feel that it has merit but does not fully meet PLOS ONE’s publication criteria as it currently stands. Therefore, we invite you to submit a revised version of the manuscript that addresses the points raised during the review process.

ACADEMIC EDITOR: **Authors are requested to remove the reviewer-suggested citations from both Version 1 and Version 2 of the manuscript. PLOS ONE does not encourage the inclusion of citations that are not directly relevant to the study. Inclusion of such citations solely at the reviewers’ request may conflict with journal policy and could negatively impact the manuscript’s evaluation, potentially leading to rejection.**

**The editor assures that removal of these suggested citations from both versions will not influence the editorial decision on the manuscript.**

We look forward to receiving your revised manuscript.

Kind regards,

Chandrabose Selvaraj, Ph.D.

Academic Editor

PLOS One

Journal Requirements:

Additional Editor Comments (if provided):

Authors are requested to remove the reviewer-suggested citations from both Version 1 and Version 2 of the manuscript. PLOS ONE does not encourage the inclusion of citations that are not directly relevant to the study. Inclusion of such citations solely at the reviewers’ request may conflict with journal policy and could negatively impact the manuscript’s evaluation, potentially leading to rejection.

The editor assures that removal of these suggested citations from both versions will not influence the editorial decision on the manuscript.

Reviewers' comments:

Reviewer's Responses to Questions

**Comments to the Author**

Reviewer #1: (No Response)

Reviewer #2: (No Response)

2. Is the manuscript technically sound, and do the data support the conclusions?

Reviewer #1: Yes

Reviewer #2: Yes

3. Has the statistical analysis been performed appropriately and rigorously?

Reviewer #1: Yes

Reviewer #2: Yes

4. Have the authors made all data underlying the findings in their manuscript fully available?

Reviewer #1: Yes

Reviewer #2: Yes

5. Is the manuscript presented in an intelligible fashion and written in standard English?

Reviewer #1: Yes

Reviewer #2: Yes

Reviewer #1: All other comments have been addressed except the raw images that yo provided have been merged together. You were asked to provide all the individual images, for each individual concentration in triplicates, that too in deperate jpg/png/tiff format with high resolution, uncropped and unedited, machine/instrument generated images. That means for each dose, there will be 3 replicate images to be seperately sent not clubbed together as you have sent the S1, S2 and S3 raw images.

Additionally, please consult and cite the following important references how to write the methods and results part of molecular docking:

a). DOI: 10.1016/j.insi.2025.100143

b). DOI: 10.1155/joch/5133015

c). DOI: 10.1016/j.micpath.2024.106627

d). From silico to benchtop: cosmosiin as a PD-1/PDL-1 immune checkpoint inhibitor revealed through DFT, network pharmacology analysis, and molecular docking integrated experimental verification.

e). Novel C-3 and C-20 derived analogs of betulinic acid as potent cytotoxic agents: design, synthesis, in vitro and in silico studies.

If possible, you should also carry out the MD SIMULATION analysis in order to validate your docking.

Reviewer #2: (No Response)

**Do you want your identity to be public for this peer review?** For information about this choice, including consent withdrawal, please see our Privacy Policy

Reviewer #1: **Yes:** DR MANZOOR AHMAD RATHER

Reviewer #2: No

---

## [Author Response · Author response to Decision Letter 2]

14 Jan 2026

Dear academic editor and reviewers,

We sincerely appreciate you for your recognition and for providing valuable suggestions that have significantly improved the quality of our manuscript [No.: PONE-D-25-45593R1]. We concur with the position that all citations should be strictly relevant to the research. Following these valuable suggestions, the detailed point-by-point response is as below.

Comment:

Authors are requested to remove the reviewer-suggested citations from both Version 1 and Version 2 of the manuscript. PLOS ONE does not encourage the inclusion of citations that are not directly relevant to the study. Inclusion of such citations solely at the reviewers’ request may conflict with journal policy and could negatively impact the manuscript’s evaluation, potentially leading to rejection.

The editor assures that removal of these suggested citations from both versions will not influence the editorial decision on the manuscript.

Response

Thank you for your kind comments on our manuscript. We fully agree with your suggestion that irrelevant citations should not be included in the manuscript. Therefore, we have re-reviewed the opinions of the reviewers and the supplementary content of our manuscript. Compared to the version 1, the second version of our manuscript has deleted 2 references and updated 16 references. Based on your reasonable suggestions, we deleted some references from the discussion section and these deleted references were marked as "[26-30]" in the second version. Since the remaining references are related to the research and they were not designated references by the reviewers, we retained some of the updated references which were used to enhance the rigor and quality of our manuscript and did not further reference the 5 designated references on molecular docking by the reviewer. We rechecked the reference list and ensured that the retained references are not withdrawn or specific papers. For your convenience, the eleven updated and retained references of the second version are listed below. If you have further suggestions on our citations, we are honored to receive your further guidance.

“14. Zucchi A, Claps F, Pastore AL, Perotti A, Biagini A, Sallicandro L, et al. Focus on the Use of Resveratrol in Bladder Cancer. Int J Mol Sci. 2023;24(5). Epub 20230226. doi: 10.3390/ijms24054562. PubMed PMID: 36901993; PubMed Central PMCID: PMCPMC10003096.

23. Katsimperis S, Tzelves L, Feretzakis G, Bellos T, Tsikopoulos I, Kostakopoulos N, et al. Circulating Tumor DNA in Muscle-Invasive Bladder Cancer: Implications for Prognosis and Treatment Personalization. Cancers (Basel). 2025;17(12). Epub 20250608. doi: 10.3390/cancers17121908. PubMed PMID: 40563559; PubMed Central PMCID: PMCPMC12190218.

24. Attanasio G, Failla M, Poidomani S, Buzzanca T, Salzano S, Zizzo M, et al. Histological and immunohistochemical approaches to molecular subtyping in muscle-invasive bladder cancer. Front Oncol. 2025;15:1546160. Epub 20250715. doi: 10.3389/fonc.2025.1546160. PubMed PMID: 40735045; PubMed Central PMCID: PMCPMC12303812.

25. Huang J, Michaud E, Shinde-Jadhav S, Fehric S, Marcq G, Mansure JJ, et al. Effects of combined radiotherapy with immune checkpoint blockade on immunological memory in luminal-like subtype murine bladder cancer model. Cancer Biol Ther. 2024;25(1):2365452. Epub 20240611. doi: 10.1080/15384047.2024.2365452. PubMed PMID: 38860746; PubMed Central PMCID: PMCPMC11174127.

31. Song B, Wang W, Tang X, Goh RMW, Thuya WL, Ho PCL, et al. Inhibitory Potential of Resveratrol in Cancer Metastasis: From Biology to Therapy. Cancers (Basel). 2023;15(10). Epub 20230514. doi: 10.3390/cancers15102758. PubMed PMID: 37345095; PubMed Central PMCID: PMCPMC10216034.

32. Jang M, Cai L, Udeani GO, Slowing KV, Thomas CF, Beecher CW, et al. Cancer chemopreventive activity of resveratrol, a natural product derived from grapes. Science. 1997;275(5297):218-20. doi: 10.1126/science.275.5297.218. PubMed PMID: 8985016.

48. Wan P, Ren Y, Deng H, Li H. CDCA4 promotes bladder cancer progression by JAK/STAT signaling pathway. J Cancer Res Clin Oncol. 2025;151(2):46. Epub 20250124. doi: 10.1007/s00432-025-06109-w. PubMed PMID: 39856473; PubMed Central PMCID: PMCPMC11762220.

49. Liu M, Guo J, Liu W, Yang Z, Yu F. Dual Targeting of Aurora-A and Bcl-xL Synergistically Reshapes the Immune Microenvironment and Induces Apoptosis in Breast Cancer. Cancer Sci. 2025;116(7):1823-35. Epub 20250330. doi: 10.1111/cas.70072. PubMed PMID: 40159464; PubMed Central PMCID: PMCPMC12210044.

50. Chen R, Zhou Z, Meng X, Lei Y, Wang Y, Wang Y. Emerging opportunities to treat drug-resistant breast cancer: Discovery of novel small-molecule inhibitors against different targets. Front Pharmacol. 2025;16:1578342. Epub 20250829. doi: 10.3389/fphar.2025.1578342. PubMed PMID: 40949156; PubMed Central PMCID: PMCPMC12425923.

51. Zhang L, Guo W, Lu N, Tian Y, Yang J, Wang L. Advances in research on unphosphorylated STAT3: A review. Medicine (Baltimore). 2025;104(30): e43476. doi: 10.1097/MD.0000000000043476. PubMed PMID: 40725945; PubMed Central PMCID: PMCPMC12303475.

52. Wang J, Liu T, Chen P, Yin D, Zhang H, Qiu X, et al. Pharmacokinetic evaluation of two oral Resveratrol formulations in a randomized, open-label, crossover study in healthy fasting subjects. Sci Rep. 2025;15(1):24515. Epub 20250708. doi: 10.1038/s41598-025-08665-0. PubMed PMID: 40628835; PubMed Central PMCID: PMCPMC12238501.”

Comment:

Reviewer #1: All other comments have been addressed except the raw images that yo provided have been merged together. You were asked to provide all the individual images, for each individual concentration in triplicates, that too in deperate jpg/png/tiff format with high resolution, uncropped and unedited, machine/instrument generated images. That means for each dose, there will be 3 replicate images to be seperately sent not clubbed together as you have sent the S1, S2 and S3 raw images.

Additionally, please consult and cite the following important references how to write the methods and results part of molecular docking:

a). DOI: 10.1016/j.insi.2025.100143

b). DOI: 10.1155/joch/5133015

c). DOI: 10.1016/j.micpath.2024.106627

d). From silico to benchtop: cosmosiin as a PD-1/PDL-1 immune checkpoint inhibitor revealed through DFT, network pharmacology analysis, and molecular docking integrated experimental verification.

e). Novel C-3 and C-20 derived analogs of betulinic acid as potent cytotoxic agents: design, synthesis, in vitro and in silico studies.

If possible, you should also carry out the MD SIMULATION analysis in order to validate your docking.

Response

Thank you for your reasonable guidance regarding the image presentation. We provide individual original figures as requested and uploaded them separately as supplementary files. We sincerely appreciate your recommendation concerning the value of Molecular Dynamics (MD) simulations to further elucidate protein-ligand dynamics and binding stability. The current study is focused on the preliminary investigation of the inhibitory mechanism of resveratrol in bladder cancer cells. The molecular docking analysis served as a bridge between our prior database screening and subsequent experimental validation, allowing us to demonstrate from multiple perspectives that resveratrol inhibits bladder cancer cell proliferation via the AURKA/STAT3 axis. We fully recognize that MD simulations can provide deeper mechanistic insights beyond static docking. At the same time, we have explained that the subsequent experiments will focus more on the construction of resveratrol stability and in vivo studies in the discussion section of our manuscript. Your valuable suggestions have provided excellent ideas for our future mechanism research. Moreover, we also thank you for providing five valuable references related to molecular docking methodology. In accordance with explicit instruction of the academic editor and journal that inclusion of specific citations solely at the reviewers’ request may conflict with journal policy, we have not included these citations in our manuscript. Nevertheless, we agree that these important references provide a very solid theoretical foundation for our subsequent exploration of the molecular dynamics between natural substances and proteins, as well as for the detection of protein stability and its interaction with ligands. We carefully read and collected these documents for continuous learning. Once again, we would like to thank you for your valuable suggestions that have helped us improve our manuscript and provided important guidance for subsequent research.

Thank you again for your valuable time and the constructive feedback. We wish you a smooth start to the new year and look forward to hearing from you regarding our manuscript.

---

## [Editor Report · Decision Letter 2]

20 Jan 2026

Resveratrol inhibits bladder cancer proliferation by targeting the AURKA/STAT3 axis: From computational analysis to experimental validation

PONE-D-25-45593R2

Dear Dr. Ren,

We’re pleased to inform you that your manuscript has been judged scientifically suitable for publication and will be formally accepted for publication once it meets all outstanding technical requirements.

Kind regards,

Chandrabose Selvaraj, Ph.D.

Academic Editor

PLOS One
---

## [Editor Report · Acceptance letter]

PONE-D-25-45593R2

PLOS One

Dear Dr. Ren,

I'm pleased to inform you that your manuscript has been deemed suitable for publication in PLOS One. Congratulations! Your manuscript is now being handed over to our production team.

Kind regards,

on behalf of

Dr. Chandrabose Selvaraj

Academic Editor

PLOS One